

# Nonlinear intensification of monsoon low pressure systems by the BSISO

Kieran M. R. Hunt[1,2] and Andrew G. Turner[1,2]

[1]Department of Meteorology, University of Reading, UK
[2]National Centre for Atmospheric Sciences, University of Reading, UK

**Correspondence:** Kieran M. R. Hunt (k.m.r.hunt@reading.ac.uk)

**Abstract.** More than half of the rainfall brought to the Indian subcontinent by the summer monsoon is associated with low-pressure systems (LPSs). Yet their relationship with the Boreal Summer Intraseasonal Oscillation (BSISO) – the dominant intraseasonal forcing on the monsoon – is only superficially understood. Using reanalysis data, we explore the relationship between the BSISO and LPS intensity, propagation, and precipitation, and associated underlying mechanisms.

The BSISO has a large impact on mean monsoon vorticity and rainfall as it moves northward – maximising both in phases 2-3 over southern India and phases 5-6 over northern India – but a much weaker relationship with total column water vapour. We present evidence that LPS genesis also preferentially follows these phases of the BSISO. We identify significant relationships between BSISO phase and LPS precipitation and propagation: for example, during BSISO phase 5, LPSs over north India produce 51% heavier rainfall and propagate northwestward 20% more quickly. Using a combination of moisture flux linearisation and quasigeostrophic theory, we show that these relationships are driven by changes to the underlying dynamics, rather than the moisture content or thermodynamic structure, of the monsoon.

Using the example of LPSs over northern India during BSISO phase 5, we show that the vertical structure of anomalous vorticity can be split into contributions from the BSISO and the nonlinear response of the LPS to anomalous BSISO circulation. Complementary hypotheses emerge about the source of this nonlinear vorticity response: nonlinear frictional convergence and secondary barotropic growth. We show that both are important. The BSISO imparts greater meridional shear on the background state, supporting LPS intensification. The BSISO background and nonlinear LPS response both contribute significantly to anomalous boundary layer convergence, and we show through vortex budget arguments that the former supports additional LPS intensification in boundary layer while the latter supports faster westward propagation.

This work therefore yields important insights into the scale interactions controlling one of the dominant synoptic systems contributing to rainfall during the monsoon.

## 1 Introduction

The summer monsoon impacts the Indian subcontinent from June to September each year, where it is responsible for 80% of the annual rainfall, totalling almost 1 m over the season (Turner and Annamalai, 2012). The summer monsoon is very heterogeneous in both space and time, being subject to significant large-scale forcing from intraseasonal oscillations that result



in extended periods of regional drought or deluges (Krishnamurthy and Shukla, 2000; Goswami and Ajayamohan, 2001; Pai et al., 2016). But the monsoon is also frequently punctuated by low-pressure systems (LPSs), of which there are usually about 15–20 per season and which are responsible for more than half of the total seasonal rainfall (Boos et al., 2015; Hunt and Fletcher, 2019).

## 1.1    The relationship between the BSISO and the South Asian summer monsoon

The Boreal Summer Intraseasonal Oscillation (BSISO) is the strongest external driver of intraseasonal variability in the summer monsoon (Kikuchi, 2021). While boreal winter is dominated by the Madden-Julian Oscillation (MJO) that propagates eastward in time, the BSISO has an additional northward component (Kikuchi et al., 2012). This means that its passage is typically associated with northwest-southeast orientated bands of alternately enhanced and suppressed convection that propagate either to the north or northeast. Like the MJO (Wheeler and Hendon, 2004), it is typically separated into eight phases, representing

octants of the phase space describing its leading modes of variability. Phase 1 is usually associated with enhanced convection over the equatorial Indian Ocean, phases 2 and 3 with enhanced convection over the Indian Ocean and East Asia, phases 4 and 5 with enhanced convection over India and the Maritime Continent, phases 6 and 7 with enhanced convection over the Bay of Bengal and South China Sea, and phase 8 with enhanced convection over the Western North Pacific (Lee et al., 2013). The average time between each phase is about 5.5 days (Lee et al., 2013).

The interaction between the BSISO and the summer monsoon is often thought to be the primary cause of so-called 'active' and 'break' monsoon periods (Wang and Xie, 1997; Webster et al., 1998). In reality, the relationship is slightly more complicated – the BSISO is described by a continuous two-dimensional phase space, whereas the active/break periods describe points on this phase space with extrems of locally enhanced/suppressed convection over the monsoon core zone of central India. Figure 1 shows the effect of the passage of the BSISO over South Asia in a novel manner. Rather than showing traditional

composites of one map for each phase, instead we attempt to relate BSISO phases to the largest anomalies on one map. This relies on the propagating nature of the BSISO, such that the largest anomalies for each phase will occur at different locations. For each gridpoint, we thus determine under which BSISO phase the (a) highest and (b) lowest monsoon rainfall occurs; the BSISO phase is indicated by the hue, while the darkness of the colour indicates the relative intensity (see figure caption). Following convention, we assign 'phase 0' to days on which the standardised magnitude of the BSISO is less than one - that is, no

appreciable BSISO is present. The BSISO clearly has a big impact on monsoon rainfall over much of the subcontinent, where particular phases are capable of increasing rainfall by more than 50%, and more than double it in some areas. The northward propagation is evident here too, while the strongest anomalies are found over the ocean and southern peninsular India. Phases 3 and 4 are associated with the greatest increase in rainfall over Sri Lanka and southern India, whereas phase 6 is associated with the greatest increase in much of the north of the country. The role of the BSISO in suppressing rainfall is less obvious,

although the rain shadow (southeast India and nearby Bay of Bengal) is strongly affected by phases 6, 7 and 8; and rainfall across the north is consistently, if weakly, suppressed in phases 2 and 3.

    Differences in long-term average rainfall must be accounted for either by changes to the available moisture – for which we use total column water vapour, shown in Fig. 2 – or the underlying dynamics – for which we use low-level relative vorticity,



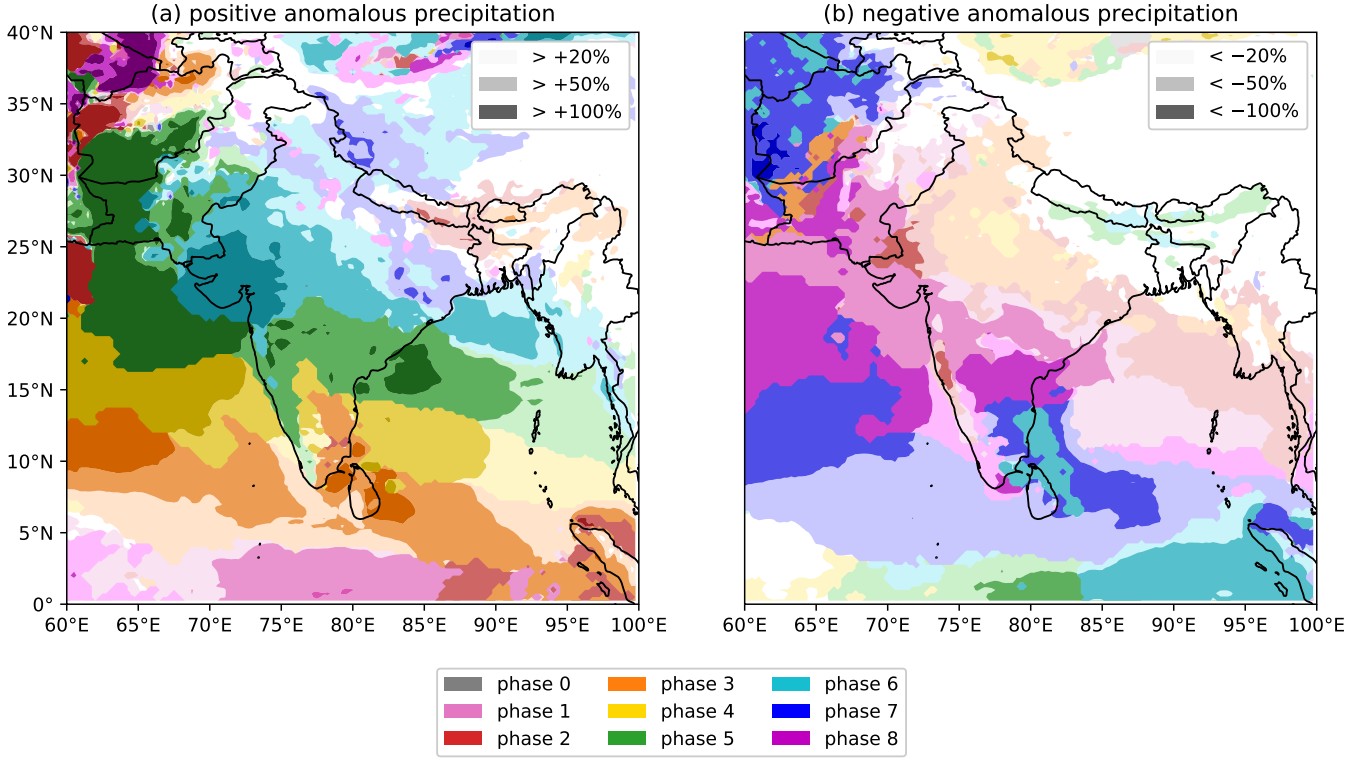

**Figure 1.** The effect of the BSISO on monsoon precipitation over South Asia. Colours indicate the BSISO phase for which mean precipitation is (a) the highest and (b) the lowest. Three levels of lightness for each colour denote the difference between the mean precipitation in that phase and the full climatology, stratified at $\pm 20\%$, $\pm 50\%$ and $\pm 100\%$.

shown in Fig. 2. Both the TCWV and vorticity have similar responses to the BSISO as the precipitation: amplification of the

climatological signal in phases 3 to 6 moving from south to north respectively, and suppression of the climatological signal by phases 6, 7 and 8 in the south and phases 2 and 3 in the north. However, there are two important differences. Firstly, the effect on TCWV is much weaker. This is partially due to thermodynamic constraints – the monsoonal lower troposphere is already nearly saturated and so can hold little additional moisture. Secondly, the effect on vorticity is almost equal and opposite when three BSISO phases apart – e.g., the region of enhanced vorticity south of India and Sri Lanka associated with phase 2 is

similarly suppressed in phase 5 – and thus the circulation response to 'active' and 'suppressed' BSISO phases is approximately symmetric. These suggest that the control exerted by the BSISO on LPSs is more likely via the monsoon circulation.

## 1.2 The relationship between monsoon LPSs and intraseasonal variability

Monsoon low-pressure systems (LPSs) are synoptic-scale circulations that typically originate over the Bay of Bengal before passing northwestward over the subcontinent. Various studies that have probed their structure using radiosondes, satellites, or

reanalyses show that they are associated with strong central vorticity, a warm-over-cold thermal structure, and a precipitation

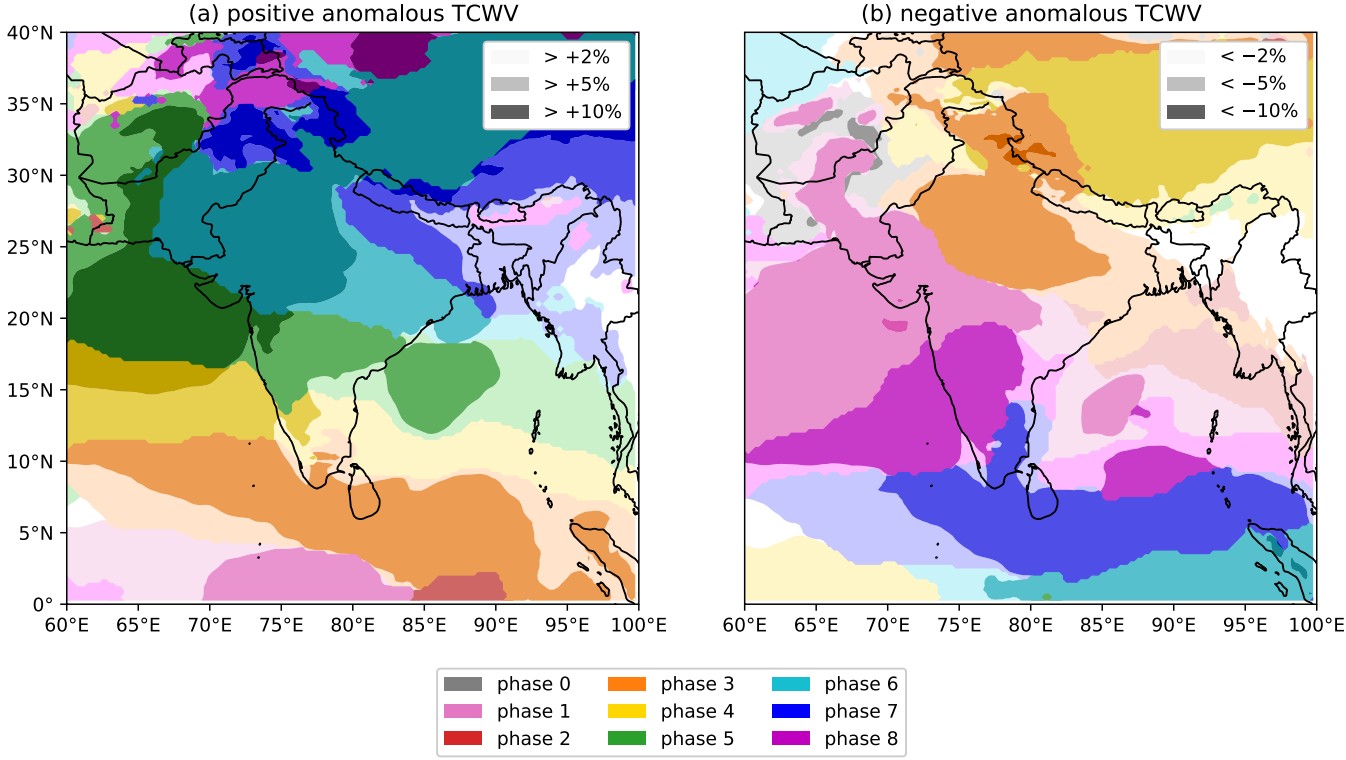

**Figure 2.** The effect of the BSISO on total column water vapour during the summer monsoon over South Asia. Colours indicate the BSISO phase for which mean TCWV is (a) the highest and (b) the lowest. Three levels of lightness for each colour denote the difference between the mean TCWV in that phase and the full climatology, stratified at $\pm 2\%$, $\pm 5\%$ and $\pm 10\%$.

maximum to the southwest of their centre (Godbole, 1977; Sarker and Choudhary, 1988; Hurley and Boos, 2015; Hunt et al., 2016). Although associated with relatively light winds, LPSs can bring extremely heavy precipitation to India, occasionally resulting in widespread flooding (Hunt and Menon, 2019; Thomas et al., 2021; Suhas et al., 2022).

Previous research on the relationship between LPSs and monsoon intraseasonal variability has often reduced the latter to
active and break phases. Such studies have invariably found that LPS activity is reduced during break phases and enhanced during active phases (e.g., Goswami et al., 2003; Krishnamurthy and Ajayamohan, 2010). More recent studies have explicitly explored the relationship between LPSs and the BSISO. Hatsuzuka and Fujinami (2017) showed that LPSs more frequently impacted Bangladesh when the BSISO was active over the northern Bay of Bengal. Karmakar et al. (2021) found that strong LPSs – known as monsoon depressions – formed more frequently when the BSISO was active over the Bay of Bengal, and
attributed this to favourable changes in large-scale vorticity and moist static energy. Deoras et al. (2021) found that the genesis of low pressure systems increased significantly over Sri Lanka when the BSISO was active there (i.e. in phases 2 and 3), and increased significantly over the Bay of Bengal as the BSISO moved northwards (i.e. into phases 4 and 5).



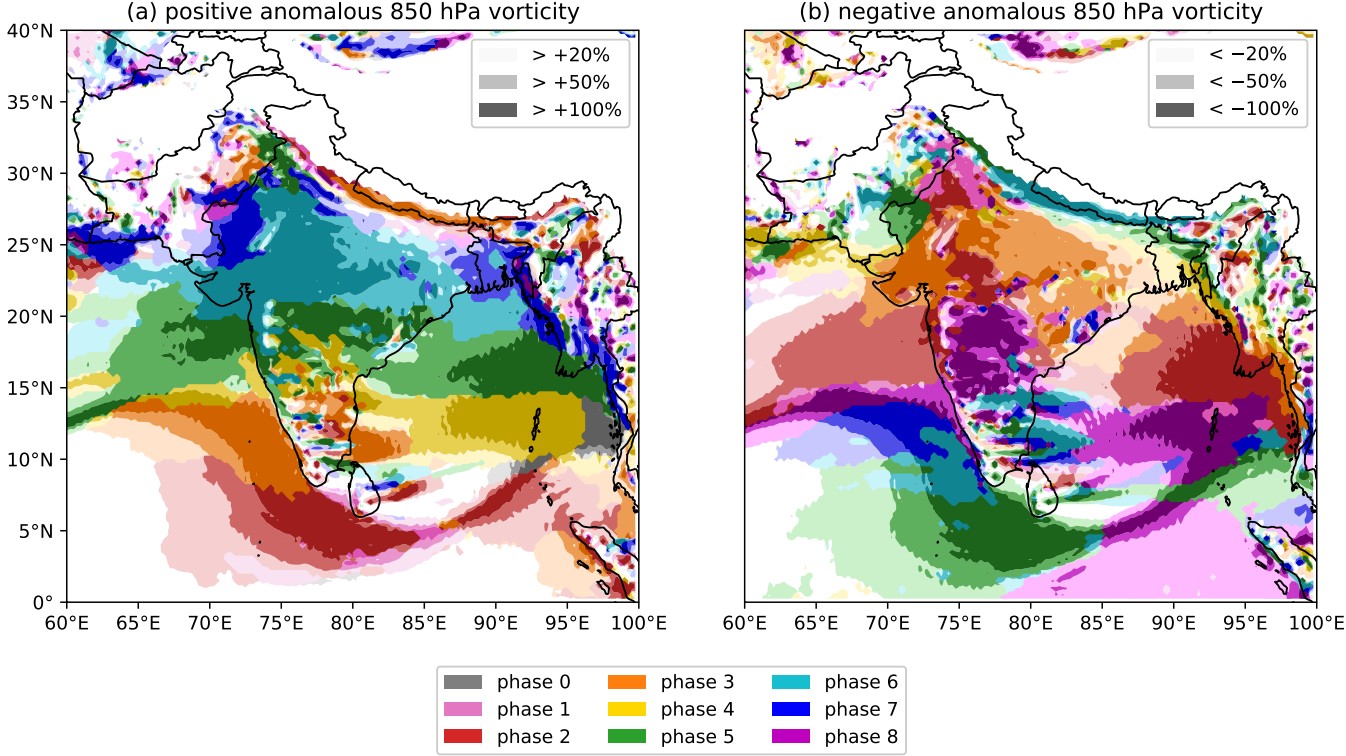

**Figure 3.** The effect of the BSISO on 850 hPa relative vorticity during the summer monsoon over South Asia. Colours indicate the BSISO phase for which mean relative vorticity is (a) the highest and (b) the lowest. Three levels of lightness for each colour denote the difference between the mean relative vorticity in that phase and the full climatology, stratified at $\pm 20\%$, $\pm 50\%$ and $\pm 100\%$. Areas where surface elevation is higher than 1500 m are masked.

So, while it is clear that the BSISO has a significant effect on LPS genesis, we still do not know how the relationship between the two affects other important LPS characteristics – such as propagation, precipitation, and intensity. These have obvious implications for LPS predictability and impacts.

This study is laid out as follows: Sec. 2 describes the data used; our results are then broken down into subsections exploring the relationship between the BSISO and LPS genesis and propagation (Sec. 3.1), LPS precipitation (Sec. 3.2), and LPS intensification (Sec. 3.3). We discuss the implications and shortcomings of our work and conclude in Sec. 4.

## 2 Data

### 2.1 ERA5

To investigate the variations in structure of the monsoon and LPSs in response to modulation by the BSISO, we use data from the ECMWF ERA5 reanalysis (Hersbach et al., 2020). Data are available globally, at hourly resolution from 1950



onwards, on a 0.25° grid. Data are available over 37 pressure levels from 1000 to 0.01 hPa, as well as at or near the surface. Data are assimilated into the forecasting system from a large variety of sources, including satellites, automatic weather

stations, and radiosondes. Data were downloaded using the dedicated API from https://cds.climate.copernicus.eu/cdsapp#!/dataset/reanalysis-era5-pressure-levels.

## 2.2 BSISO indices

Data for the BSISO were downloaded from http://iprc.soest.hawaii.edu/users/kazuyosh/Bimodal_ISO.html, whose methodology is described in Lee et al. (2013). It is used to assess the role of large-scale intraseasonal variability in modulating the

diurnal cycle of precipitation over the Himalayas. The dataset has daily resolution, available from 1979–2018. It contains the normalised values of the first two principal components (or RMMs), as well as the resulting phase and amplitude.

## 2.3 LPS track dataset

We use the database of low-pressure system (LPS) tracks from Hunt and Fletcher (2019) in this study. Using six-hourly ERA-Interim data, they tracked LPSs by computing the mean relative vorticity in the 900-800 hPa layer, then performing a spectral

truncation at T63 to filter out short-wavelength noise. They then identified regions of positive relative vorticity within this field and determined the centroid location for each one. These centroids were then linked in time, subject to constraints in distance and steering winds, to form candidate LPS tracks. This algorithm has been used for monsoon LPSs by a number of authors (e.g. Martin et al., 2020; Dong et al., 2020; Roy and Rao, 2022). Track data are available at https://doi.org/10.5281/zenodo.5575336. For this study, we only consider LPSs that spin up between June 1 and September 30, leaving a total sample of 885 tracks from

1979 to 2018.

## 3 Results

We first consider the effect of the BSISO on general LPS genesis and propagation behaviour (Sec. 3.1), before examining the effect on LPS precipitation and the role played by quasigeostrophic and diabatic processes (Sec. 3.2) and finally the role of nonlinear interactions (Sec. 3.3).

### 3.1 Effect of the BSISO on LPS genesis and propagation

As discussed in the introduction, while recent studies have shown a strong relationship between locally active BSISO conditions and LPS genesis frequency (Hatsuzuka and Fujinami, 2017; Karmakar et al., 2021; Deoras et al., 2021), they have typically been limited to regional analysis. We saw in Figs. 2 and 3 how large-scale variables associated with the monsoon respond dynamically to the northward propagation of the BSISO. These variables – low-level vorticity and total column water vapour

– have also been shown to be important predictors of the genesis of stronger monsoon depressions (Ditchek et al., 2016), and so we hypothesise that the broader class of LPSs responds in a similar way.



**Figure 4.** The effect of the BSISO on monsoon LPS genesis. LPS genesis points (Jun–Sep 1979–2018) are coloured according to the phase of the BSISO in which they occurred. The background field estimates these values over a 1° grid by averaging all points using an inverse-square distance weighting. Genesis points occurring when the amplitude of the BSISO is less than one – about 41% of the total sample – are not included.





We test this two ways in Fig. 4. Firstly, we plot the genesis locations of all 523 LPSs that spun up during a defined phase of the BSISO (i.e., in which the amplitude exceeds one), colouring those points by the active phase at the time. Secondly, to test coherence, we convert these points into a gridded field: for each point $(i,j)$ on a $1° \times 1°$ grid, the mean phase, $\overline{\phi_{ij}}$ is computed
over all genesis points, weighted by the inverse square of their distances to that gridpoint, while taking into account the fact that BSISO phase is a circular variable, i.e.:

$$\overline{\phi_{ij}} = \lfloor (\overline{\Phi_{ij}} \mod 8) \rfloor, \tag{1}$$

where

$$\overline{\Phi_{ij}} = \frac{4}{\pi} \mathrm{atan}\left( \frac{\Sigma_p w_{pij} \sin(\phi_p \cdot \pi/4)}{\Sigma_p w_{pij}}, \frac{\Sigma_p w_{pij} \cos(\phi_p \cdot \pi/4)}{\Sigma_p w_{pij}} \right), \tag{2}$$

where $\phi_p$ is the BSISO phase associated with LPS $p$, and the weights, $w_{pij}$, are given by:

$$w_{pij} = (d(i,j,x_p,y_p))^{-2}. \tag{3}$$

Here, $d(x_1,y_1,x_2,y_2)$ is a function describing the geodesic separation between two longitude-latitude pairs, $(x_1,y_1)$ and $(x_2,y_2)$, and the subscript $p$ indicates individual LPS genesis points which are to be summed over in Eq. 2.

We see a strong relationship between local BSISO phase and LPS genesis that responds to the northward propagation of
the BSISO. LPSs most commonly form over Sri Lanka in phases 1 and 2, LPSs forming during phase 4 predominate over the central Bay of Bengal where the mean LPS genesis rate is slightly lower, and in phase 5 in a thin band north of that. Across northern India and the head of the Bay of Bengal, the picture is more mixed but phases 6, 7 and 8 are evidently the most favourable for genesis. Overall, the mean genesis location for LPSs in each of the eight phases are all significantly different from each other. These results are in agreement with earlier studies, but place the relationship into a larger, and spatial, context.
We also note that the phases associated with LPS genesis are roughly half a phase behind those linked with enhanced low-level vorticity in Fig. 3. We attribute this to the delay between the BSISO producing a favourable environment and the convective response to that environment organising onto a scale that is large enough to reach geostrophic balance and spin up into LPSs. Taking into account all the analysis so far, the general sequence of events is an increase in vorticity (e.g. during phase 1 over Sri Lanka), followed by an increase in LPS genesis (e.g. during phases 1 and 2 over Sri Lanka), followed by increased TCWV
and precipitation (e.g. during phase 3 over Sri Lanka).

The remaining 362 LPSs that occur during 'phase 0' – i.e., when the BSISO amplitude is insufficient for the phase to be well defined – share a very similar distribution to the 523 plotted in Fig. 4. This suggests an insignificant relationship between BSISO amplitude and LPS genesis, in agreement with similar results for the MJO (Haertel and Boos, 2017).

As the BSISO significantly affects the large-scale environment in which LPSs are embedded, including moist static energy
(Karmakar et al., 2021) and mid-level steering winds (Hsu et al., 2017), we may also expect it to affect LPS propagation, for which the steering winds are a particularly important factor (Boos et al., 2015). To test this hypothesis, we separate LPSs into those either side of 15°N – roughly demarcating those forming over or near Sri Lanka and along the southern flank of the monsoonal westerlies from those forming over the head of the Bay of Bengal or within the monsoon trough – since the



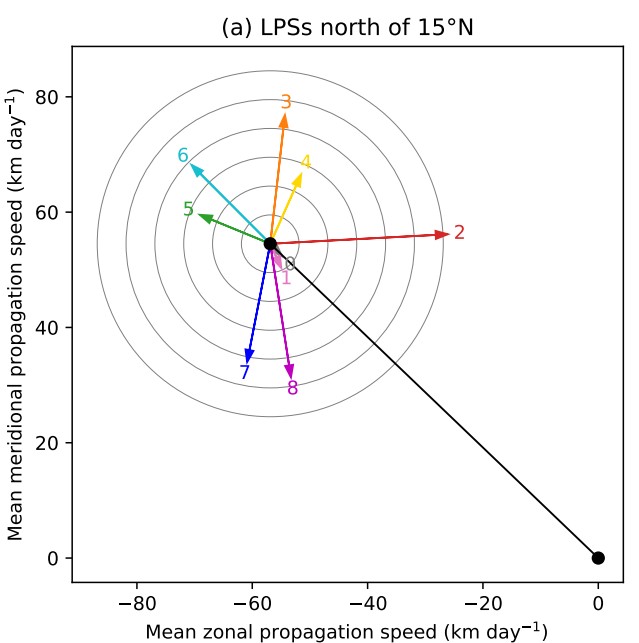

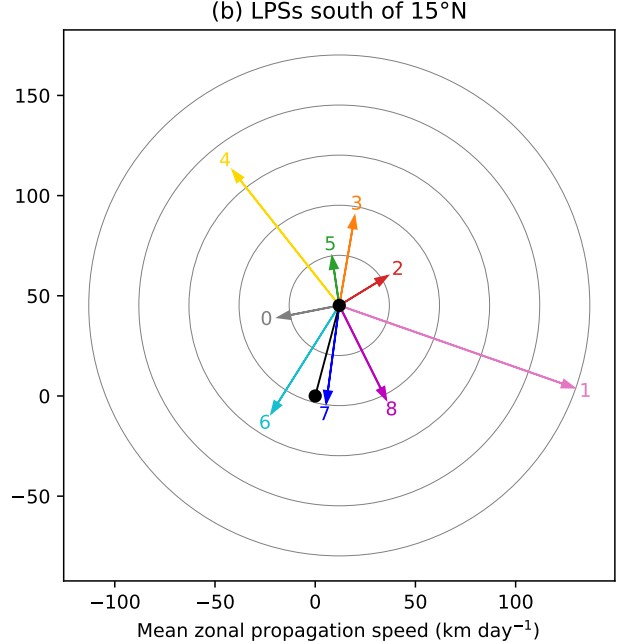

**Figure 5.** The effect of the BSISO on monsoon LPS propagation velocity for (a) systems north of 15°N and (b) systems south of 15°N. In each case, the climatological propagation velocity is given in black, with the anomalies to this mean for each BSISO phase given by coloured arrows. Rings indicate anomalous speeds in multiples of 5 and 25 km day$^{-1}$ in (a) and (b) respectively.

climatological circulation and effect of the BSISO differ considerably in each region. Based on the 500 hPa geopotential
composites presented in Hsu et al. (2017), as well as additional 500 hPa wind composites (not shown), we would expect strong
anomalous westward transport of LPSs north of 15°N in phases 5 and 6, and strong anomalous eastward transport in phases 8,
1 and 2. Similarly, for systems south of 15°N, we would expect strong anomalous westward transport in phases 1 and 2, and
strong anomalous eastward transport in phases 5 and 6.

To measure this in more detail, Fig. 5 shows the mean and anomalous propagation vectors for LPSs on either side 15°N.
For those north of 15°N, the mean propagation vector for each BSISO phase except 1 (and 0) differs significantly from the
overall mean. We see some of the expected features: anomalous west/northwest propagation during phases 5 and 6, amounting
to an increase in the mean propagation speed of about 13% and 22% respectively. Similarly, anomalous eastward propagation
in phase 2 slows the mean propagation speed down by about 20%. However, the anomalous northward propagation in phase 3
and southward propagation in phases 7 and 8 cannot be explained by changes in the 500 hPa winds, implying that some other
mechanism is at play.

The relationship breaks down completely for LPSs south of 15°N, where the BSISO has an even stronger effect on propagation velocity. During phase 1, which is associated with reasonably strong anomalous mid-level easterlies over Sri Lanka, LPSs
have a large eastward component to their anomalous (and even total) propagation vector – i.e. the wrong direction altogether.
During phase 5, which is associated with the strongest anomalous mid-level westerlies, LPSs have a small anomalous north-





ward component to their propagation. It is clear that although LPSs may generally be steered by 500 hPa winds, the effect of
the BSISO on their propagation does not operate by this mechanism.

## 3.2 Effect of the BSISO on LPS precipitation



**Figure 6.** The effect of the BSISO on precipitation in monsoon LPSs north of 15°N. The central panel shows the mean composite system-centered precipitation for all such LPSs. Anomalies to this mean as a function of BSISO phase, along with their respective sample sizes (total six-hourly timesteps), are given in the surrounding panels. Precipitation data from ERA5.

As we have seen, passage of an active BSISO event can have a significant impact on local rainfall (Fig. 1) over monsoonal
India. Since the majority of monsoon rainfall is brought by LPSs (Hunt and Fletcher, 2019), we ask whether it is simply the

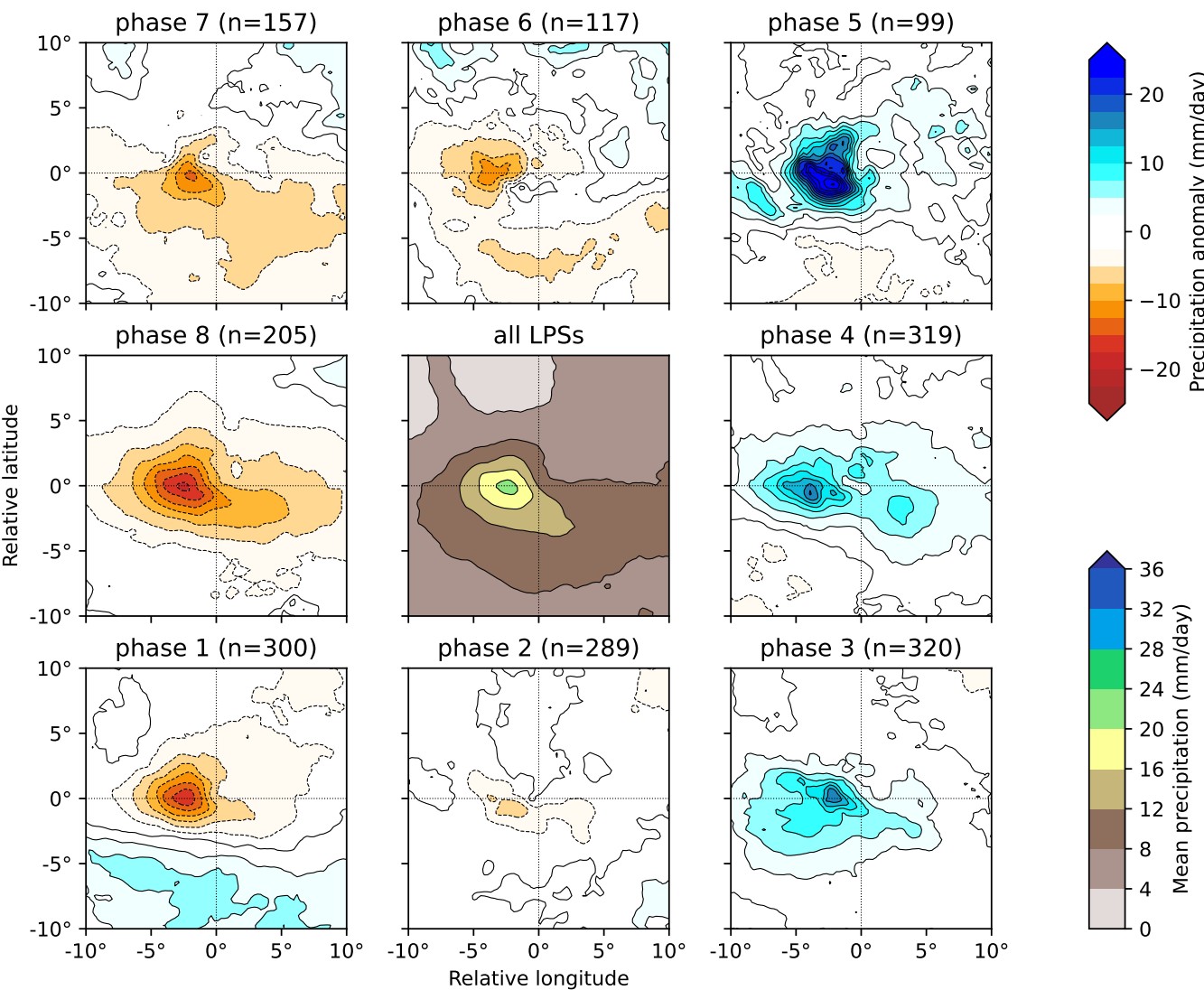

**Figure 7.** The effect of the BSISO on precipitation in monsoon LPSs south of 15°N. The central panel shows the mean composite system-centered precipitation for all such LPSs. Anomalies to this mean as a function of BSISO phase, along with their respective sample sizes, are given in the surrounding panels. Precipitation data from ERA5.





case that the BSISO just results in more LPSs (as in Fig. 4), or whether some additional interaction between the two is resulting in a change in LPS rainfall. Here, we again separate our analysis into LPSs on either side of 15°N. For each group, we plot the mean system-centered precipitation in the centre and use the eight surrounding subplots to show the anomaly to this mean for each of the eight defined BSISO phases. These are ordered counterclockwise from bottom left, consistent with depictions of MJO and BSISO phase space in previous studies (Wheeler and Hendon, 2004; Lee et al., 2013).

Results for LPSs north of 15°N are given in Fig. 6. Each phase has a good sample size, though there are about twice as many LPS timesteps in phase 5 as in phase 1. Phases 1, 2 and 3 result in a significant decrease in LPS precipitation – reducing the composite maximum by 27%, 39% and 19% (and overall means by 13%, 15% and 1%) respectively. Phases 5, 6 and 7 result in a significant increase in LPS precipitation – increasing the composite maximum by 51%, 35% and 14% (and overall means by 11%, 12% and 5%) respectively. These changes are coherent over the BSISO cycle, rising from a single minimum in phase 2 to

a single maximum in phase 5. Notably, the rainfall difference imparted by any given phase is not spatially uniform – instead it is closely related to the existing spatial distribution of LPS rainfall. This suggests that it is not just a case of the BSISO creating an environment with a relatively uniform increase in rainfall through which LPSs simply pass or interact weakly. Rather, the BSISO somehow affects the mechanisms driving LPS rainfall.

A similar picture emerges for LPSs south of 15°N (Fig. 7), though rotated two phases earlier. The mean rainfall associated

with these LPSs is substantially lower than their northern counterparts, but the absolute change in rainfall associated with different BSISO phases is similar. This results in much larger fractional changes, not unlike those we saw for propagation earlier. Phases 7, 8 and 1 result in a significant decrease in LPS precipitation – reducing the composite maximum by 32%, 48% and 27% (and overall means by 35%, 45%, and 6%) respectively. Phases 5, 6 and 7 result in a significant increase in LPS precipitation – increasing the composite maximum by 76%, 68% and 146% (and overall means by 25%, 27% and 25%)

respectively.

Let us now investigate why these anomalies occur – i.e. what does the BSISO do to LPSs to suppress or enhance their precipitation? For the sake of brevity, we will now restrict our analysis to consider only LPSs north of 15°N, and only to the two BSISO phases – 2 and 5 respectively – that most strongly suppress and enhance precipitation in such LPSs. Here, we can use moisture flux convergence to understand the synoptic-scale changes responsible for influencing precipitation, since the two

fields are strongly related in the tropics (e.g. Neelin and Held, 1987).

Fig. 8 shows vertically integrated moisture flux and moisture flux convergence, both as composite means for all monsoon LPSs north of 15°N, and as anomalies to these means for LPSs in BSISO phases 2 and 5. The composite mean (Fig. 8(a)) shares many similarities with LPS-centered moisture flux composites from earlier studies (Hurley and Boos, 2015; Hunt et al., 2016; Deoras et al., 2022): cyclonic but with substantial meridional shear, and with a convergence maximum several hundred

kilometres to the southwest – colocated with the precipitation maximum.

The anomalous fields (Figs. 8(b) and (c)) show a reduction in both cyclonicity and meridional shear for LPSs in phase 2, and an increase in both for LPSs in phase 5, with the associated moisture flux convergence behaving accordingly in each case. We also see anomalies of the opposite sign between 700 and 900 km north of the centre in both phases, suggesting either the



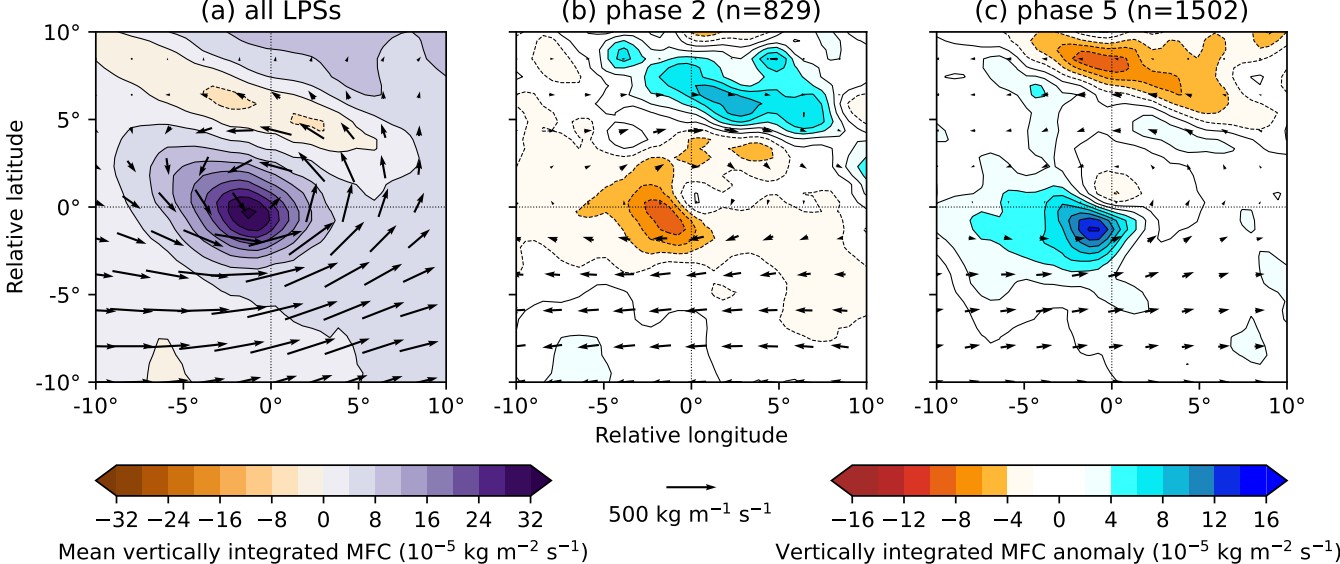

**Figure 8.** The effect of the BSISO on vertically integrated hPa moisture flux and moisture flux convergence in monsoon LPSs north of 15°N. The composite for all such LPSs is given in (a), with the differences for LPS composites from BSISO phases 2 and 5 given in (b) and (c) respectively, along with their sample sizes.

development of an anomalous secondary meridional circulation within the LPS or an additional interaction between the BSISO

and the Himalayan foothills to the north. The foothills start at about 26°N, consistent with the composite anomalies.

We also see, particularly in phase 5, that the anomalous increase in moisture flux convergence is not exactly collocated with the maximum in the full composite – instead being further south. The implication of this is that the anomalous moisture flux convergence does not explain the changes in maximum precipitation well: in phase 5 the maximum convergence increases by 37% (cf. 51% for precipitation), and in phase 2 the maximum convergence decreases by 24% (cf. 39% for precipitation).

Changes in moisture flux (and hence moisture flux convergence) can be understood through a simple linearisation:

$$(q\mathbf{v})' - \overline{q\mathbf{v}} = \underbrace{q'\overline{\mathbf{v}}}_{\text{moisture}} + \underbrace{\overline{q}\mathbf{v}'}_{\text{wind}} + \underbrace{q'\mathbf{v}'}_{\text{nonlinear}}, \tag{4}$$

where barred quantities indicate the all-LPS mean and primed quantities indicate the anomaly to that mean for the phase of interest. The first term on the right hand side denotes the contribution to the anomalous moisture flux from changes to moisture alone, given in the left hand panels of Fig. 9. This is approximately an order of magnitude smaller than the second term on the

right hand side – the contribution from anomalous winds alone – which is given in the right hand panel of Fig. 9. The nonlinear third term, the product of the two anomalies, is smaller still and neglected here. Note that for simplicity, and because the two fields are highly congruent, we use 850 hPa moisture flux here rather than the full vertical integral.

Fig. 9 clearly indicates that increased moisture flux and moisture flux convergence associated with LPSs in BSISO phase 5 (and reduced flux and convergence in phase 2) are, to first order, due to changes in the circulation rather than the moisture field.



**Figure 9.** Contribution to LPS-centered anomalous 850 hPa moisture flux and moisture flux convergence in (a) BSISO phase 2 and (b) BSISO phase 5 from (left) anomalous humidity and (right) anomalous circulation. In each case the contribution is computed by linearising about the mean LPS composite. Computed for monsoon LPSs north of 15°N.





This is perhaps to be expected, given that we saw in Figs. 2 and 3 that the BSISO had a much greater effect on vorticity than
total column water vapour over the monsoon. As we will discuss shortly, there is still a potential secondary role for moisture,
in providing perturbations to mid- and upper-level latent heating, modulating low-level ascent.

Vertical velocity ($\omega$) is diagnostically linked to low-level convergence through the continuity equation and so we examine it
here. The composite mean 750 hPa $\omega$ and the respective BSISO phase 2 and 5 anomalies are shown in Fig. 10. As expected,
the mean closely resembles the composite precipitation and moisture flux convergence fields with a strong maximum in ascent
a few hundred kilometres southwest of the system centre. The anomalies for phases 2 and 5 (Figs. 10(b) and (c) respectively)
share a very similar form, though with opposite sign. The minimum in $\omega$ (maximum in ascent) is reduced in magnitude by 37%
in phase 2 and increased by 54% in phase 5, closely following the changes in precipitation (39% and 51% respectively).

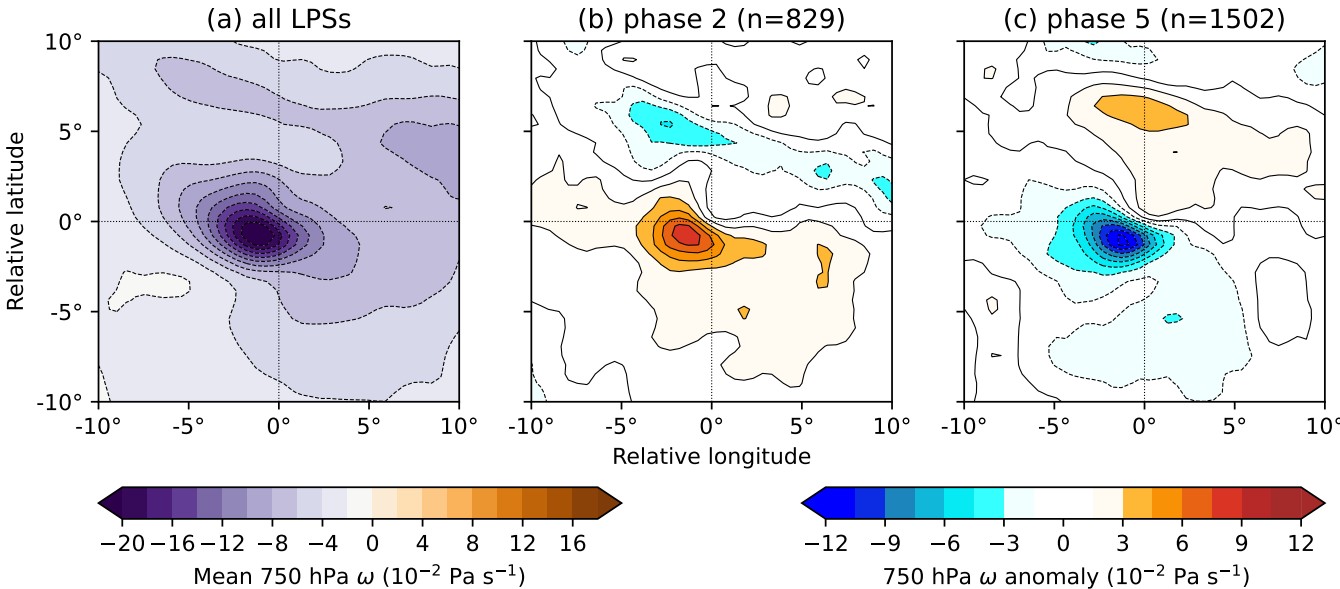

**Figure 10.** The effect of the BSISO on 750 hPa $\omega$ (vertical velocity) in monsoon LPSs north of 15°N. The composite for all such LPSs is
given in (a), with the differences for LPS composites from BSISO phases 2 and 5 given in (b) and (c) respectively, along with their sample
sizes. Points where the surface pressure is less than 750 hPa are not included in the composites. Following convention, negative values of $\omega$
indicate upward motion.

We can further break down the changes in vertical velocity using the (also diagnostic) $\omega$-equation:

$$\left(\nabla_p^2 + \frac{f}{\sigma}\frac{\partial^2}{\partial p^2}\right)\omega = \underbrace{\frac{f_0}{\sigma}\frac{\partial}{\partial p}[\mathbf{v}_g \cdot \nabla_p(\zeta_g + f)] + \frac{R}{\sigma p}\nabla_p^2[\mathbf{v}_g \cdot \nabla_p T]}_{\text{quasigeostrophic forcing}} + \underbrace{\frac{R}{c_p\sigma p}\nabla_p^2 Q}_{\text{local diabatic heating}}, \tag{5}$$

where the symbols have their usual meanings. Here, the left hand side can be thought of as a three-dimensional Laplacian of
vertical velocity, with the operator behaving effectively like a negative sign. The first term on the right hand side represents
forcing due to differential advection of geostrophic absolute vorticity, the second term represents the forcing due to temperature

## (a) quasigeostrophic ω forcing

**Figure 11.** Mean system-centered $\omega$ forcing at 750 hPa, split into (a) quasigeostrophic and (b) diabatic contributions. Left: the composite mean for all monsoon LPSs north of 15°N; centre: the difference between all LPSs and those occurring during BSISO phase 2; right: the difference between all LPSs and those occurring during BSISO phase 5.





advection, and the third term represents forcing due to local diabatic heating. The first two terms comprise the quasigeostrophic

forcing, which alone explain the existence of an ascent maximum – and hence precipitation maximum – in the southwest quadrant of LPSs (Rao and Rajamani, 1970). We neglect forcing due to friction, as it is negligible above the boundary layer (e.g. Bluestein, 1993).

We can now use the $\omega$-equation in Eq. 5 to explain these differences, splitting the right-hand side into two parts: the quasigeostrophic component, which explains the forcing due to synoptic-scale dynamics; and the final term, which explains forcing

due to gradients in diabatic heating. We include this final term for completeness, even though it is typically small for monsoon LPSs (Boos et al., 2015), because it may play an important role in secondary circulations (Chen et al., 2005).

These two terms are shown in Fig. 11(a) and (b) respectively for the composite of all LPSs north of 15°N, and for phase 2 and phase 5 anomalies. The quasigeostrophic forcing explains the $\omega$ pattern reasonably well, as expected from previous studies (Sanders, 1984; Boos et al., 2015), and is more than an order of magnitude stronger than the diabatic forcing – in both the mean

and anomalies. This difference in the anomalies is in agreement with the relative difference in mean vorticity and total column water vapour perturbations caused by the BSISO (Figs. 2 and 3).

Breaking the quasigeostrophic forcing into its two components (not shown), we note that the differential vorticity term easily dominates the temperature advection term. So, it is clear that when the BSISO affects LPS intensity and precipitation, either enhancing them (phases 5–7 for systems north of 15°N) or weakening them (phases 1–3), it does so through modulating

the circulation of the monsoon rather than its moisture content or thermodynamic structure. Mean and anomalous forcing at 400 hPa, approximately the level of the upper-tropospheric warm core, is similarly balanced but about five times weaker than at 750 hPa (not shown). In the next section, we will explore precisely how the interaction between LPSs and the perturbed monsoon lead to changes in LPS intensity and propagation.

### 3.3   Nonlinear intensification

We start by familiarising ourselves with the vertical structures of relevant LPS fields – vertical velocity and relative vorticity – and comparing these to the anomalies for LPSs during BSISO phase 5 (Fig. 12). Note that for the sake of brevity, analysis will now be constrained to BSISO phase 5, which features the strongest positive perturbation to LPS precipitation. Throughout, unless stated otherwise, the results for phase 2 can be taken to be approximately equal and opposite.

The vertical structure of composite monsoon LPSs has been the subject of many studies (e.g. Godbole, 1977; Hurley and

Boos, 2015) and so we need only note the salient features here. Vertical velocity has a maximum at about 750 hPa, roughly 100 km west of the centre. Ascent is deep, effectively reaching the tropopause, but is very weak in the boundary layer. Vorticity has a maximum at 850 hPa, in the centre, with positive values extending up to about 300 hPa. Like ascent, it is weaker in the boundary layer, though not to the same extent.

Anomalous ascent in phase 5 has a very similar pattern to the mean, which it increases by an average of about 20% over the

cross-section. This corroborates the $\omega$-forcing analysis in the previous section (Fig. 11), where we showed that the perturbation to quasigeostrophic forcing generally acted multiplicatively. This is not the case for anomalous vorticity (Fig. 12(d)). Here, the changes do not appear to act on the mean structure, and comprise two components. Firstly, a broad, relatively weak increase





**Figure 12.** Zonal-vertical cross-section through system-centred composite means of $\omega$ (a,b) and relative vorticity (c,d). For each variable, the mean for all LPSs is given on the left (a,c), and the anomaly for LPSs occurring during BSISO phase 5 is given on the right (b,d). Computed for monsoon LPSs north of 15°N. Cross-sections are taken as an average from 1° south to 1° of the central latitude. Data from levels below surface pressure have been excluded.

that is largely uniform throughout most of the troposphere, and is stronger towards the west of the system; and secondly, a much smaller-scale, much larger increase, confined to the centre and mostly embedded in the boundary layer.

These changes can be explained by taking the anomaly in Fig. 12(d) and splitting it into contributions from the BSISO background state and the subsequent nonlinear LPS response to that state, respectively, shown in Fig. 13. To obtain the BSISO background contribution, we compute the mean three-dimensional anomalous relative vorticity over India and the surrounding region during BSISO phase 5, and then sample this field using the relevant LPS tracks as we would for a normal composite. To limit the effect of monsoon onset and withdrawal, the anomalies are computed using monthly means; e.g. the anomaly for 26

June 2016 is computed using June mean vorticity. The nonlinear effect (panel (c)) is then taken as the difference between the actual anomaly (panel (a)) and the BSISO background effect (panel (b)).



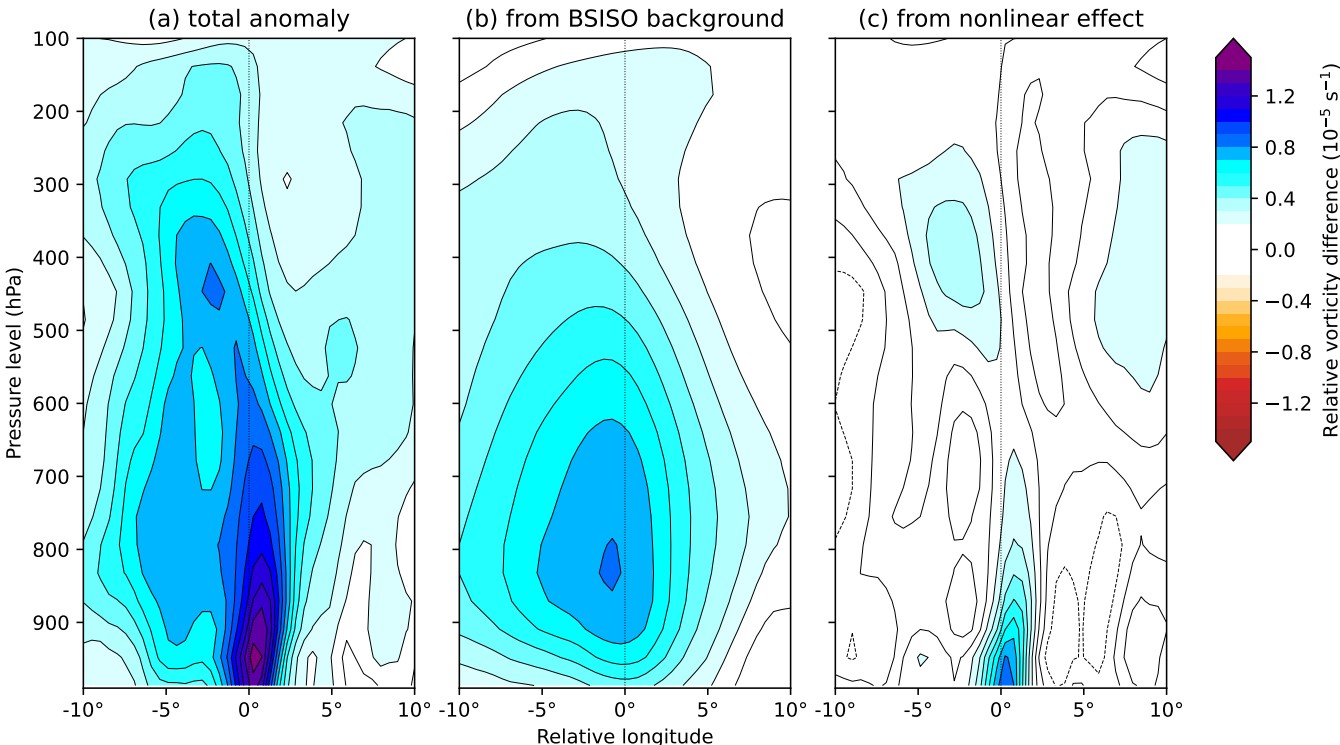

**Figure 13.** Zonal-vertical cross-sections of anomalous system-centred relative vorticity for LPSs in BSISO phase 5, compared with the all-LPS mean. The total anomaly is given in (a), the contribution from the BSISO alone is given in (b), and the contribution from the interaction between the BSISO and LPSs is given in (c). Computed for monsoon LPSs north of 15°N.

The background effect of the BSISO is to impart the broad and relatively weak anomaly extending throughout the troposphere that we identified in Fig. 12. This represents an intensification of the monsoon trough, through which we can explain its structure: its greatest effect just above the boundary layer, at about 850 hPa, where monsoon winds are typically strongest, and the asymmetry favouring higher values towards the west reflects the fact that climatological (and hence anomalous) monsoon vorticity is greater over the peninsula than over the Bay of Bengal.

The nonlinear term, i.e. the contribution to the anomalous vorticity due to the interaction between the LPS and the BSISO, is given in Fig. 15. The anomalous vorticity is mostly confined to the boundary layer and very near to the system centre, where it reaches a comparatively large magnitude. There are two potential causes of this. Firstly, LPSs are thought to grow through moist barotropic instability, extracting energy from the basic state of the monsoon (Diaz and Boos, 2019). As phase 5 increases the intensity of the monsoon background state, LPSs could extract proportionally more energy and thus intensify further than in the absence of that perturbation. Secondly, frictional convergence scales approximately quadratically with near surface winds (Smagorinsky, 1963), therefore an LPS vortex and a vortical BSISO background state constructively interfere





to produce frictional convergence – and hence, through stretching, relative vorticity in the boundary layer – that is greater than
the sum of its parts.

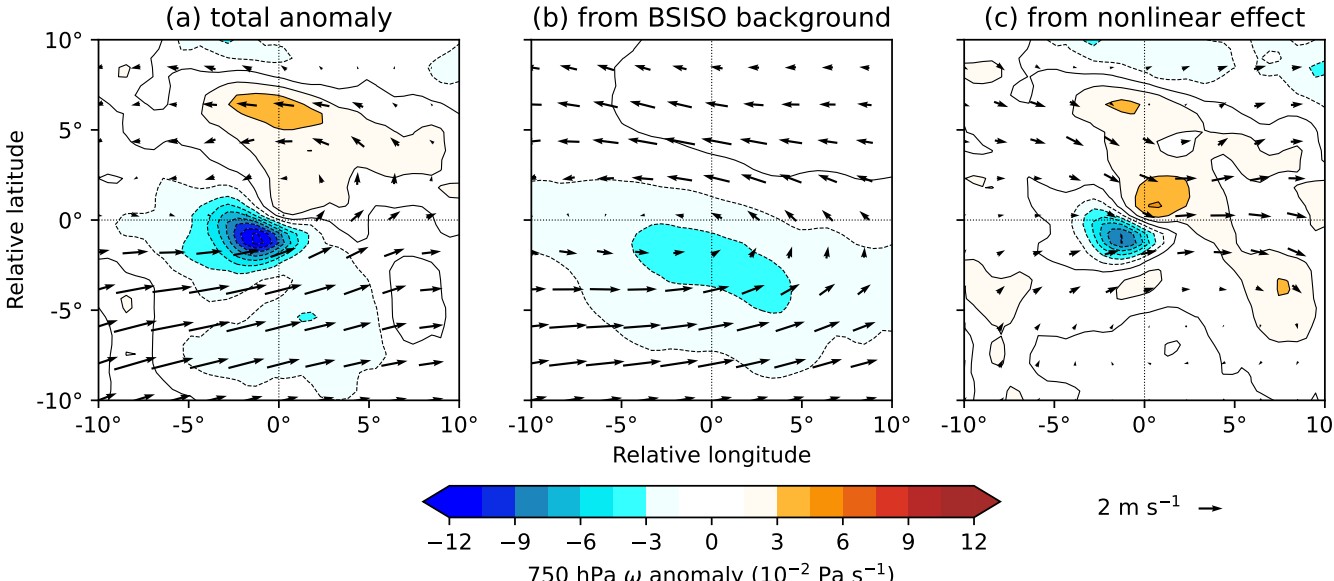

**Figure 14.** Anomalous system-centred 750 hPa $\omega$ and winds for LPSs in BSISO phase 5, compared with the all-LPS mean. The total anomaly is given in (a), the contribution from the BSISO alone is given in (b), and the contribution from the interaction between the BSISO and LPSs is given in (c). Computed for monsoon LPSs north of 15°N.

To test the first of these hypotheses, we split the composite anomaly of both horizontal and vertical winds using the same technique as in Fig. 13. Fig. 14 shows a horizontal cross-section through this composite taken at 750 hPa. The BSISO background state imparts a relatively strong cyclonic anomaly at this level, centred slightly to the south of the composite LPS. This is associated with a weak but relatively large-scale increase in ascent, located further south. In contrast, the nonlinear

contribution explains most of the anomalous ascent; but it is the winds that are of particular interest here. The presence of the LPS reduces the meridional shear associated with the BSISO, as easterlies present to the north of the composite LPS in the background state are significantly reduced in magnitude by the nonlinear term. This implies that LPSs in this region can indeed extract the additional shear present during phase 5 of the BSISO, consequently intensifying more through moist barotropic instability than they would otherwise.

This only explains some of the anomalous vorticity in Fig. 13, since the anomalous winds associated with the BSISO background state are relatively weak in the boundary layer and barotropic growth is not sufficiently strong there to counteract that (Diaz and Boos, 2019). We therefore investigate the second of our hypotheses: the contribution of nonlinear frictional convergence. Vertical composites of convergence, zonal, and vertical winds are given in Fig. 15, separated in the same way as in Figs. 13 and 14.



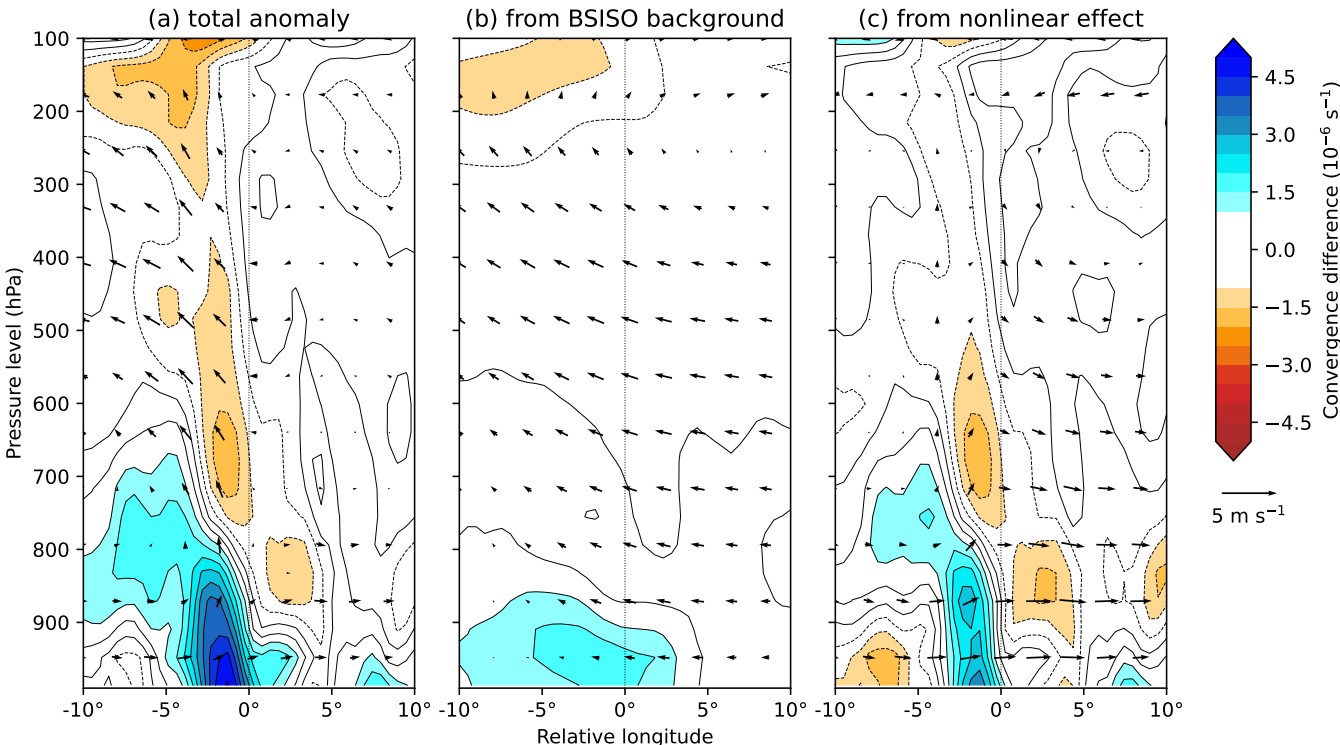

**Figure 15.** Zonal-vertical cross-sections of anomalous system-centred convergence and winds (parallel to the plane, i.e., $u$ and $w$ components, with $w$ exaggerated by a factor of 200, approximately in keeping with the figure aspect ratio) for LPSs in BSISO phase 5, compared with the all-LPS mean. The total anomaly is given in (a), the contribution from the BSISO alone is given in (b), and the contribution from the interaction between the BSISO and LPSs is given in (c). Computed for monsoon LPSs north of 15°N.

The anomalous convergence for LPSs in BSISO phase 5 is largest in the boundary layer, particularly near the surface, and is confined to within a few hundred kilometres of the centre of the LPS. The contribution of the anomalous BSISO background state to this is – as with vorticity – relatively broad, and, although positive at the centre of the LPS, biased towards the west. The nonlinear contribution, caused by the interaction between the composite LPS and the anomalous BSISO background state, explains most of the total anomaly. However, it is located away from the centre, meaning that it is unlikely to be directly

responsible for the increase in vorticity seen in Fig. 13(c), since the resultant stretching would occur several hundred kilometres too far west.

    This out-of-phase relationship between convergence and vorticity – known as moisture vortex instability – has been observed in monsoon LPSs (Adames and Ming, 2018). However, it is unlikely to be occurring here since it requires the long timescales provided by organised deep convection, whereas this interaction deals with shallow convection confined largely to the boundary

layer. However, the convergence caused by the BSISO background state is also reasonably strong: the composite maximum of $1.3 \times 10^{-6}$ s$^{-1}$ is only about 30% weaker than the nonlinear convergence maximum ($1.9 \times 10^{-6}$ s$^{-1}$), and so may play an important role in vortex stretching.



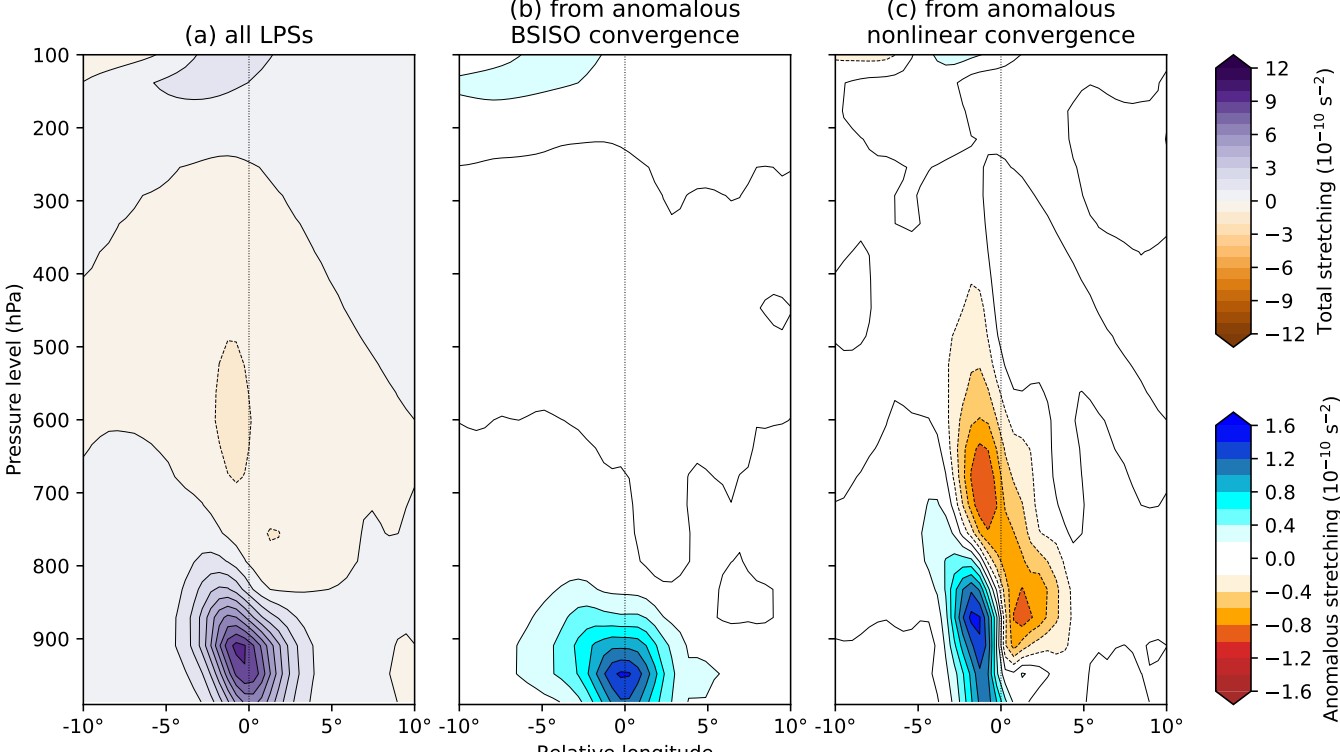

**Figure 16.** Zonal-vertical cross-sections of vortex stretching in monsoon LPSs. The composite average for all LPSs is given in (a). The remaining subplots show contributions to the anomalous stretching in LPSs during BSISO phase 5 from (b) convergence from the mean BSISO itself and (c) convergence provided by the nonlinear interaction between LPSs and the BSISO. Computed for monsoon LPSs north of 15°N.

The vertical structure of anomalous vortex stretching ($\zeta \nabla \cdot \mathbf{u}_\mathrm{h}$) is shown in Fig. 16. Here, we show the mean vortex stretching for all LPSs north of 15°N (panel (a)), and the anomalous terms ((b) and (c)) represent the stretching imparted on phase-5 LPSs by the BSISO background state and nonlinear LPS convergence respectively. Two contrasting patterns emerge. The convergence provided by the BSISO stretches the vortex most strongly at the LPS centre and in the boundary layer – this pattern is similar to the overall mean, but about 15% of the magnitude, and will result in boundary layer and low-level intensification of the LPS, in addition to that provided further aloft by the increased barotropic instability discussed in the analysis of Fig. 10.

On the other hand, stretching provided by the nonlinear convergence occurs off-centre, to the west, and extends more deeply into the lower troposphere, where it peaks at about 850 hPa, the typical level associated with maximum LPS vorticity (Fig. 12(c)). This has implications for propagation – increased stretching to the west results in an increase in $\partial\zeta/\partial t$ there, accelerating westward LPS movement during BSISO phase 5. LPSs are typically thought to propagate through advection by mid-level steering winds (Boos et al., 2015) and interaction with the orography to the north (Hunt and Parker, 2016), but we have seen that anomalous steering winds do not explain the anomalous LPS propagation seen during different BSISO





phases, and of course the orography is fixed. Therefore, even though vortex stretching plays a relatively minor role in mean LPS propagation – notably it is largest to the southwest (Sanders, 1984; Chen et al., 2005) although LPSs propagate to the northwest – we propose that it might at least partially explain the anomalous propagation in Fig. 5.

## 4 Concluding remarks

In this study, we investigated the relationship between the BSISO – the largest mode of intraseasonal variability affecting the
South Asian monsoon – and low-pressure systems – the dominant synoptic event responsible for monsoon rainfall.

We started by quantifying the effect of the BSISO on the mean monsoon, where it has a strong relationship with both low-level vorticity and rainfall – in some regions more than doubling their climatological values – with the effect moving northwards from phases 2 and 3 over the south of the peninsula to phases 5 and 6 over the monsoon trough and Himalayan foothills. The 'active' (i.e. where rainfall and vorticity are enhanced) and 'break' (i.e. where rainfall and vorticity are suppressed) phases of
the BSISO behave largely in an equal and opposite manner.

We showed that LPS genesis similarly follows this northward propagation of BSISO activity, reporting results similar to Deoras et al. (2021), that LPS genesis is strongly preferred over Sri Lanka in phase 2, over the central Bay of Bengal in phase 4, and over the head of the Bay of Bengal in phases 5 and 6. This relationship is strongest in regions with climatologically low monsoon LPS genesis, like the Arabian Sea and the centre and south of the Bay of Bengal. This relationship formed part
of a delayed relationship: BSISO activity first increases the background relative vorticity, then, after a few days, LPS genesis increases, followed by total column water vapour and precipitation. Further research is still needed to better understand the relationship between favourable conditions, organised convection, and LPS genesis. We found no relationship between BSISO amplitude and LPS genesis, following similar results reported for the MJO by Haertel and Boos (2017).

Before moving onto composite analysis, we split LPSs into two groups – north and south of 15°N – noting that the two
subsets exist in significantly different environments, affected by different phases of the BSISO.

We found that the phase of the BSISO strongly affects the propagation of LPSs in both groups. Northern LPSs travel northwestward about 20% faster during phases 5 and 6, but about 20% slower in phase 2. Southern LPSs are much more strongly affected, with their propagation direction – not just speed – being a function of BSISO phase. For example, mean propagation is southeast in phases 8 and 1, northeast in phases 2 and 3, northwest in phases 4 and 5, and southwest in phases
6 and 7. We argue that even though mean propagation is well explained for all LPSs by mid-level steering winds (as in Boos et al., 2015), these propagation anomalies are not.

We also found that the phase of the BSISO strongly affects LPS precipitation. Rainfall associated with northern LPSs is significantly increased during BSISO phases 5, 6 and 7, and significantly reduced during phases 1, 2 and 3. This has a profound impact on the composite rainfall maximum – which is 51% higher than the all-LPS mean during phase 5 and 39% lower than
the all-LPS mean during phase 2. For southern LPSs, even though their mean rainfall is much lower, the absolute effect of the BSISO is comparable to that for northern LPSs, although this is shifted through two phases on account of their different mean





latitudes so that rainfall is increased during phases 3, 4 and 5 and reduced during phases 7, 8 and 1. This results in much larger fractional changes: the composite maximum is more than twice as high during phase 5 as in the all-LPS mean.

By linearising the anomalous moisture flux convergence associated with LPSs during phase 5, we showed that the large rainfall anomalies are explained by changes imparted by the BSISO on monsoon circulation, rather than moisture. Using the $\omega$-forcing equation and quasigeostrophic theory, we further narrow this down to the result of changes to the vertical structure of vorticity of the monsoon by the BSISO, as opposed to changes in the thermodynamic structure or large-scale diabatic heating from additional convection.

 We then used the case of northern LPSs occurring during BSISO phase 5 to investigate the how the BSISO affects internal
LPS dynamics. We found that the vertical structure of anomalous vorticity – that is the difference in vorticity between phase 5 LPSs and all LPSs – could be separated into two distinct components: that provided directly by the BSISO, and that arising from a nonlinear interaction between the LPS and the BSISO. The background BSISO term is broad in scale and persists through much of the troposphere, but is relative weak; in contrast, the nonlinear term is strongly confined to the centre of the LPS where it is strongest in the boundary layer and does not penetrate much into the troposphere. We suggested two hypotheses
for the origin of this nonlinear term. Firstly, that it arises due to the approximately quadratic relationship between near-surface winds and frictional convergence (Smagorinsky, 1963); and secondly, that the LPS is responding to the increased barotropic instability of the background state.

 We found that both hypotheses were valid, with the barotropic instability argument likely to explain increased vorticity in the lower troposphere, and the frictional convergence argument likely to explain the increased vorticity in the boundary layer.
Similarly separating anomalous convergence into BSISO and nonlinear contributions, we showed that both were important in the boundary layer, and that the BSISO contribution was in fact only slightly weaker. Future work could employ targeted modelling to understand the relationship between and relative importance of these two sources of convergence and low-level vorticity.

 Finally, we computed anomalous vortex stretching terms by multiplying these anomalous convergence terms with the total
vorticity. The convergence associated with the BSISO background state is mostly centred on the LPS, as is its related stretching, which we suggest results in increased LPS intensity. In contrast, the stretching associated with the nonlinear term occurs off-centre, to the west, where we argue it supports the anomalously fast northwestward propagation of phase 5 LPSs. Further exploration of the role of the interaction between anomalous vortex stretching and anomalous steering winds in driving anomalous LPS propagation in different BSISO phases is left for future work.

*Code availability.* All code for analysis and plotting to be made available on GitHub repository. All data used are freely available with repository links given in Sec. 2.





*Author contributions.* KMRH performed the data analysis and led the conceptual design and drafting of the manuscript following discussion with AGT. The manuscript was iterated by AGT, who also procured the funding for the work.

*Competing interests.* The authors declare no conflict of interest.

*Acknowledgements.* KMRH and AGT are funded through the Weather and Climate Science for Service Partnership (WCSSP) India, a collaborative initiative between the Met Office, supported by the UK Government's Newton Fund, and the Indian Ministry of Earth Sciences (MoES).



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
