# Peer review of "Nonlinear intensification of monsoon low pressure systems by the BSISO"

_Weather and Climate Dynamics, 2022_

## Author Response (AR1)

**REVIEWER 1**

This manuscript tries to understand a long-standing problem in the context of the Indian monsoon: the relationship between intraseasonal oscillations and the generation of low-pressure systems. Few recent studies have shown the association, but here the authors provide a detailed overview and try to understand how LPS are modulated by ISO phases in both rainfall intensity and propagation speed. Dynamical and thermodynamical aspects are investigated and the authors concluded that the nonlinear frictional convergence and anomalous boundary layer convergence both are important for LPS intensification. This work has immense scientific importance in understanding monsoon dynamics. I enjoyed reading it! There are a few points that still need to be clarified:

1. Line 5: Provide some explanation about the phases before this statement. How do you get the phases?
I think given that the BSISO is a well-documented phenomenon in tropical meteorology, the abstract is probably not a suitable place to include this level of detail. However, following this suggestion, we have now expanded the description in the Data section: "In short, empirical orthogonal function analysis is applied to outgoing longwave radiation and 850 hPa zonal wind fields over a region covering most of South and Southeast Asia (40-160°E, 10°S-40°N). The first two principal components then describe a phase space that describes the northward propagating variability often linked with MJO activity. This phase space is described in polar coordinates, giving the BSISO an amplitude and a phase. Although the phase is a continuous circular variable, it is typically discretised into integers 1-8.".

2. Line 47: The results presented in Figures 1-3 are interesting. While you discussed data and methods later in Section 2, putting these figures in the introduction does not tell the source of the dataset. I suggest moving these figures to the main Results and keeping a discussion based on earlier studies in the introduction.
We have made this change – pushing the three introductory figures and related discussion into a new Section 3.1 "Effect of the BSISO on the mean monsoon".

3. Line 59: Fig. 3 instead of Fig. 2.
Corrected. Thank you.

4. Lines 109-110: Which region did you choose for the analysis?
We have updated the LPS data section to include this information: "For this study, we consider only those LPSs that impact the Indian subcontinent, rejecting tracks that do not pass through [70-90°E, 4-24°N]. We also only consider LPSs that spin up during the summer monsoon, i.e., between June 1 and September 30. This leaves a total sample of 885 tracks from 1979 to 2018."

5. Figure 4: What is the meaning of the color over the grid points where there are no LPS formed during the period of analysis (e.g., near the Himalayan foothills or the western equatorial Indian Ocean)?

This is an inverse-square-weighted circular mean, and so is defined everywhere. However, we agree that to colour grid squares far away from any LPS genesis is meaningless and have updated the figure to reflect that. Our new figure (below) colours squares white if there is no LPS genesis within 300 km during the study period.

[Figure]

6. Line 140: What do you mean by "half a phase"?

We have added a footnote here to explain: "Recall that BSISO phase numbers are discretised from a continuous circular distribution, with e.g., 'phase 3' including phase numbers between 2.5 and 3.5. In the discrete space used in our maps, half a phase is the difference between being in the spatial centre of a phase and being at its boundary."

7. Line 143: I could see both phases 1 and 2

We agree and have amended the wording to reflect this as follows: "Taking into account all the analysis so far, the general sequence of events is an increase in vorticity, simultaneous with an increase in LPS genesis (e.g. during phases 1 and 2 over Sri Lanka), followed by increased TCWV and precipitation (e.g. during phase 3 over Sri Lanka)."

8. Lines 146-147: How do you conclude this if the "amplitude is insufficient for the phase to be well defined"? Did you disregard the amplitude and considered just the phase in these cases?
We did not phrase this clearly. What we meant here was that the overall distributions of both sets (amp<1, amp>1) of LPS genesis points is similar. In other words, both sets possess maxima over the head of the Bay of Bengal and Sri Lanka, with minorities over the Indo-Gangetic Plain, Arabian Sea, and central Bay of Bengal. We make no phase-related claim here. The wording has been revised to reflect this: "The remaining 362 LPSs that spin up during 'phase 0' – i.e., when the BSISO amplitude is insufficient for the phase to be well defined – have genesis points distributed similarly to the 523 plotted in Fig. 4, i.e., maxima at the head of the Bay of Bengal and over Sri Lanka with additional contributions from the Indo-Gangetic Plain, Arabian Sea, and central Bay of Bengal."

9. Figure 5 and text: How do you calculate LPS propagation velocity?
This is calculated as the displacement vector between successive timesteps divided by the length of the timestep itself. The propagation vectors are then averaged according to the phase in which they occur. We have added a short sentence at the top of the paragraph explaining this: "The vectors are computed for each LPS timestep and then grouped according to the BSISO phase in which they occur."

10. Lines 160-165: The large-scale circulation during the genesis day of the LPS could be very different when the LPS is matured (maybe 4-5 days later). How do you interpret the results? This analysis is based on the genesis days as I understand it.
No, the analysis is based on simultaneous days. We have now clarified this at the top of the section by adding "The vectors are computed for each LPS timestep and then grouped according to the BSISO phase in which they occur." Even so, the BSISO has a period of 30-60 days, meaning it usually sits in a given phase for 4-8 days.

11. Figure 6: Is this precipitation on genesis day or averaged over the lifespan of the LPS? If it is the later case as it appears looking into the number of samples, how do you consider the BSISO phase as LPS life may span across multiple BSISO phases?
Neither. As with propagation, the composites are based on simultaneous BSISO phase, to ensure a fair comparison. For example, consider a five-day LPS, for which the BSISO phases on each day are 4 4 4 5 0. Precipitation from the first three days goes into the composite for phase 4, precipitation from the fourth day goes into the phase 5 composite, and data from the fifth day are excluded. We have added the following text to clarify this: "As with propagation (and hereafter), compositing is done according to simultaneous BSISO phase, meaning a single LPS may be distributed across several composites."

12. Figure 6: "all such LPSs" - including those in phase 0?
Yes. We have updated the caption to reflect this.

13. Lines 190-195: Phase 5 in both cases increases the anomaly in rainfall: this could be related to the fact that LPS may have a lifespan across different BSISO phases. As we clarified in our response to point #11, the composites are constructed using simultaneous, rather than genesis, BSISO phases. Therefore, although this is a good suggestion, it is not valid here. If I had to speculate, I would suggest that during phase 5, based on Figure 2, there is anomalous TCWV in a band (stretching across central and southern India) that both northern and southern LPSs can access.

14. Lines 275-280: Did you do the compositing based on BSISO phase 5 days without any LPS activity and with LPS activity, and saw the differences? This is correct, except the background includes all days (not just non-LPS days, although they comprise the majority). As we explain in the original text: "To obtain the BSISO background contribution, we compute the mean three-dimensional anomalous relative vorticity over India and the surrounding region during BSISO phase 5, and then sample this field using the relevant LPS tracks as we would for a normal composite. To limit the effect of monsoon onset and withdrawal, the anomalies are computed using monthly means; e.g. the anomaly for 26 June 2016 is computed using June mean vorticity."

15. Line 285: this is rather difficult to conclude here. Agreed, we have replaced this with a generalisation: "This represents an intensification of the monsoon trough, through which we can explain its structure: its greatest effect just above the boundary layer (at about 850 hPa, where monsoon winds are typically strongest) and the asymmetry favouring higher values towards the west reflects the fact that LPSs are more commonly found in the eastern part of the monsoon trough."

Looking forward to the answers and revised manuscript.

**REVIEWER 2**

Monsoon low pressure systems (LPS) are responsible for a major part of summer monsoon rainfall over the Indian subcontinent and are also linked with extreme precipitation. This timely paper investigates the relationship between monsoon LPS and Boreal summer intraseasonal oscillations (BSISO), one of the dominant modes of intraseasonal variability affecting the South Asian monsoon. The authors show in detail the way BSISO affects the LPS genesis and precipitation. They find the modulation of LPS by BSISO occurs through dynamic rather than thermodynamic changes. Using the QG-omega equation they show that the anomalous vorticity changes are driven mostly by the differential vorticity advection and further explore how the various factors affect the structure of anomalous vorticity. Overall, I find the results are sufficently novel and of great scientific importance. I have a few clarifications/corrections listed below, and I'd be happy to recommend it for publication once these points are addressed.

Comments:

L12-13: The phrase "Contributions from BSISO and anomalous BSISO circulations " is confusing. Better to rephrase contributions from BSISO as from the "mean/background" BSISO circulation.
Agreed, we have made this change so that it now reads: "we show that the vertical structure of anomalous vorticity can be split into contributions from the BSISO background circulation and the nonlinear response of the LPS to anomalous BSISO circulation".

L37: Do phases 4 and 5 bring enhanced convection over all of India or just a specific region there?
Figure R2.1 below shows OLR anomalies (W m$^{-2}$) by BSISO phase, from Kikuchi (2021; doi: 10.2151/jmsj.2021-045). We see that phase 4 brings enhanced convection to much of southern India, whereas phase 5 brings enhanced convection to central and parts of north India. So, although it has a large footprint, the anomalous convection associated with these phases does not cover the whole of India.

[Figure]

*Figure R2.1*

Liked this novel way of presenting different phases in Fig1. Does individual maps of precipitation for various phases show similar behaviour of northward propagation of the precipitation peaks.

Thank you. To satisfy the reviewer's curiosity, we include Figure R2.2 below showing anomalous monsoon precipitation over India during each of the BSISO phases. The northward propagation of both positive and negative anomalies as a function of phase is clearly visible.

[Figure]

*Figure R2.2*

I would suggest moving the figures 1-3 from introduction to results section.
We have made this change – pushing the three introductory figures and related discussion into a new Section 3.1 "Effect of the BSISO on the mean monsoon".

L59-65: I know it's been cited earlier in the introduction but it might be a good thing to mention the earlier studies that show intensification of background vorticity related to BSISO here.
We have added "This is the same pattern reported in Kikuchi et al. (2012) and Lee et al. (2013)." To the end of this paragraph.

Fig 4: Would it be meaningful to color the grid boxes with no LPS?
This depends on the box in question. We would argue that a box with LPSs on either side, but which contains none itself, can be appropriately coloured (i.e. interpolation). However, colouring those outside the overall genesis region (i.e. extrapolation) is indeed questionable. Therefore, in the revised manuscript, we replace this figure with one where boxes are not coloured if there is no LPS within 300 km. The figure (Fig R2.3) is attached below.

[Figure]

*Figure R2.3*

How sensitive is the classification with the grid sizes and the classification criteria. It might be better to use k-means clustering or may be a simple criterion like the phase to which the majority of LPS points within some radial distance belong to. These are good questions which merit further exploration. Firstly, we show sensitivity to grid spacing in Figure R2.4 below. Given our use of an inverse square weighted mean results in a continuously defined field, all we do by changing the grid spacing is alter how regularly that field is sampled. There are, therefore, no significant changes between the choices of 0.5°, 1° and 2°.

[Figure]

Figure R2.4

The next part of the comment asks about making the appropriate choice of region classifier, suggesting *k*-means or some variant of nearest neighbour. *k*-means would not be appropriate here as it a clustering algorithm, not a classification algorithm. There is a problem with standard classification algorithms in general here: they are designed to work with discrete, categorical variables. Instead, we have a variable that is defined continuously over a circle (only discretised later by convention). This means that classifiers take phase 4 to be as different from phase 3 as it is from

phase 8. This problem does not affect our algorithm as we perform appropriate circular decomposition first.

Despite this potential shortcoming, we test several classifiers below (Figure R2.5). The two best performing of these, yielding similar results to each other (and to our own result) are the nearest-neighbour algorithm and the support vector classification (which we use here with a Gaussian kernel). Our own method is approximately a combination of these, except our kernel is an inverse square and the classification handles phase information appropriately. For these reasons, we make no changes to the manuscript.

[Figure]

*Figure R2.5*

L135: Lots of phase 7 and 8 points are also there over Sri Lanka.
Agreed. We have altered the text to reflect this: "LPSs most commonly form over Sri Lanka in phases 7, 8, 1 and 2…"

L140: It's unclear what half phase means.

We have added a footnote here to explain: "Recall that BSISO phase numbers are discretised from a continuous circular distribution, with e.g., 'phase 3' including phase numbers between 2.5 and 3.5. In the discrete space used in our maps, half a phase is the difference between being in the spatial centre of a phase and being at its boundary."

L144: Any reason why increased TCWV would follow LPS genesis. Wouldn't TCWV be a factor determining the LPS genesis?

These are not mutually exclusive, as increased TCWV may support LPS genesis, and then be further increased through the actions of the LPS (increased upward surface latent heat flux, large-scale vertical moisture flux). We show this using a lagged composite in Figure R2.6 below. For each LPS, we take a 5° box centred on the genesis point and composite mean TCWV over the box from five days before genesis until ten days after. These are additionally subset into generally active BSISO (phases 1-8) and BSISO active over the Bay of Bengal (phases 4-6). In all cases, the TCWV rises before LPS genesis, but does not reach a peak until one day after. We clarify this in the revised manuscript: "Taking into account all the analysis so far, the general sequence of events is an increase in vorticity, simultaneous with an increase in LPS genesis (e.g. during phases 1 and 2 over Sri Lanka), followed by increased TCWV – as the LPS increases upwards surface latent heat flux and vertical moisture transport – and precipitation (e.g. during phase 3 over Sri Lanka)."

[Figure]

*Figure R2.6*

Figure 6: Are these the precipitation plots for all the LPS times and not just for the genesis? If so, how are the phases assigned for the later stages of LPS. Is it based on the phase at the time of genesis or does it change with time along the LPS track?

The composites are based on simultaneous BSISO phase (i.e., varying with time along the track), to ensure a fair comparison. For example, consider a five-day LPS, where the BSISO phases on each day are 4 4 4 5 0. Precipitation from the first three days goes into the composite for phase 4, precipitation from the fourth day goes into the phase 5 composite, and data from the fifth day are excluded. We have added the following text to clarify this: "As with propagation (and hereafter), compositing is done according to simultaneous BSISO phase, meaning a single LPS may be distributed across several composites."

Figure 6 and in other places too : What does "all such LPSs" refer to, does it include phase 0?
Yes. We have updated the captions to reflect this.

L186-188: What would be the nature of the changes in the background environment that result in uniform increase in rainfall. For eg, even a uniform increase in moisture would result in precipitation increasing in the region of convergence but not a uniform increase everywhere.
Yes – we could have worded this better. We have replaced "relatively uniform" with "large-scale".

L189: "rotated two phases earlier", the maxima is still at phase 5 isn't it? Only the negative anomalies shift. Any reasons for that.
Yes, the phase with the largest anomalous precipitation is phase 5 in both cases. Here, however, we are talking about the overall pattern. The statement holds because, as we explain in the text, positive anomalies move from (5, 6, 7) in northern LPSs to (3, 4, 5) in southern LPSs and negative anomalies move from (1, 2, 3) to (7, 8, 1). We do not know why phase 5 produces the largest anomalies in both groups, but one hypothesis is that during phase 5, based on Figure 2, there is anomalous TCWV in a band (stretching across central and southern India) that both northern and southern LPSs can access. No change has been made here.

L268: Wouldn't vorticity always have a maxima in the centre as you are tracking the LPS using vorticity maxima.
Yes. We have rephrased this to put emphasis on the maximum being at 850 hPa rather than at the storm centre, replacing "vorticity has a maximum at 850 hPa, in the centre, with positive values extending up to about 300 hPa" with "vorticity has a central maximum at 850 hPa, with positive values extending up to about 300 hPa"

L285-286: Can this be deduced from the storm centre analysis alone.
Probably not. Following a similar comment from reviewer 1, we have replaced this with a generalisation: "This represents an intensification of the monsoon trough, through which we can explain its structure: its greatest effect just above the

boundary layer, at about 850 hPa, where monsoon winds are typically strongest, and the asymmetry favouring higher values towards the west reflects the fact that LPSs are more commonly found in the eastern part of the monsoon trough."

L288: Fig13 instead of 15.
Thank you – this should have been Fig 13(c).

Since it's the phase rather than the amplitude of BSISO that's affecting the LPS modulation, I'm just curious on how the background states vary for phase 0. Are the background conditions similar across the phases irrespective of the amplitude?
This is a good question. However, let us first clarify that the only claim we make regarding amplitude is that it does not have a significant relationship with LPS genesis. Still, it is worth demonstrating that "phase 0" can be safely rejected from the analysis. We show the anomalous background vorticity for each of the eight phases below, both when the amplitude is greater than one (as used conventionally and in the paper, Fig R2.7) and when it is less than one (Fig R2.8). We see that changes imparted onto the mean vorticity are 5-10 times weaker in the latter case. We also briefly investigate the link with LPSs, separating LPS-centred anomalous precipitation during BSISO phase 5 by BSISO amplitude (a<1, 1<a<2, a>2) in Fig R2.9. Again, we see the effect of the BSISO during "phase 0", i.e., when the amplitude is less than one, is insignificant. However, we also note that there is a positive correlation between anomalous precipitation and BSISO amplitude, as we might expect from the strengthening of the background shown in Figs R2.7 and R2.8. We hope this has satisfied the reviewer's curiosity!

BSISO amplitude > 1

[Figure]

Figure R2.7

**BSISO amplitude < 1**

[Figure]

*Figure R2.8*

[Figure]

*Figure R2.9*

Typos :

L43 : "extrems"
Corrected.

**REVIEWER 3**

This paper proposed to investigate the relationship between the Boreal Summer Intraseasonal Oscillation and the monsoon low-pressure systems, which is a novel topic with respect to the dynamics near the Indian ocean. The authors studied the precipitation, water vapor, vorticity and the SLP genesis during all eight phases of BSISO and selected phases 2 and 5 as representatives of opposing phases. The energetics was then illustrated through linearization and omega function based on the two typical phases. The nonlinear features in the vertical velocity and vorticity within the SLPs were highlighted by subtracting the linear part from the full fields.

The manuscript is nicely drafted but still has some problems that need to be fixed before publishing

Major opinions:

1. Are Figures 1,2 and 3 colorblind-friendly? I know you are trying to use rainbow colors to represent different phases but should be careful with the selection of colors. It looks a bit messy especially when you assign three levels of brightness to each color.
This is a fair comment from the reviewer, and clearly any figure using a wide range of hues will not be accessible to sufferers of dichromacy or monochromacy. However, we do note that for sufferers of protanomaly – the most common form of colour blindness (anomalous or weak trichromacy in the red cones) – Figs 1-3 are still readable. This is because the hue degeneracy occurs only between phases 7 and 8, as we show in Fig R3.1 (below) by applying a colour-blindness projection to our original Fig 3.

[Figure]

*Figure R3.10*

Here are two potential options to better show these plots:

1). Instead of using darkness or brightness to represent the intensity, try using contours with different line types, for example, the one on the left, use dashed contours for >20%, solid contours for 100% and no contours for 50%

We test this suggestion in Fig R3.2 below, which is analogous to Fig 3 in our original paper. We believe this makes the plots less clear since line contours do not contain directional information, meaning the reader does not know which side of each solid line is less than or greater than a ±100% anomaly. We do not, therefore, adopt this suggestion.

[Figure]

Figure R3.11

2). Separate each plot into three according to the intensity.
We test this suggestion in Fig R3.3 below, which is again analogous to Fig 3 in our original paper. Presenting the three thresholds in different plots makes intercomparison more difficult and would massively inflate the number of (sub)figures we use in Figs 1-3 from six to eighteen. For these reasons, we do not adopt this suggestion.

[Figure]

Anomalous 850 hPa relative vorticity by BSISO phase

*Figure R3.12*

Instead, we suggest a compromise, allowing us to keep the number of subfigures in the paper low but also to provide a friendly option for colour blindness. For each of Figures 1-3, we will provide a supplementary figure that decomposes the anomalies by phase (following the style used in our original Figs 6 and 7). An example of this construction is given (again for Fig 3) in Fig R3.4.

[Figure]

*Figure R3.13*

2. In line 34, you stated that BSISO is identical to MJO.
   This is not true. In line 34, we stated "Like the MJO (Wheeler and Hendon, 2004), it is typically separated into eight phases, representing octants of the phase space describing its leading modes of variability." While the BSISO and MJO share many similarities, here we only inform the reader that they are often represented in the same way (i.e. using a Wheeler-Hendon plot, or using a 1-8 integer phase numbering system).

   The phases of MJO are usually shown by the outgoing long-wave radiation, how about BSISO? Is it often shown by OLR or precipitation or any other fields? Can you make a schematic plot to show the eight phases? An amplitude-latitude or an amplitude-time or a time-latitude diagram is suggested when introducing the eight phases.
   Figures 1-3 serve as our introductory plots, concisely showing the

development of the BSISO (more-or-less as time-latitude maps) and its impacts over monsoonal South Asia. I don't really see the benefit to the reader of recycling this information into a fourth introductory plot, especially as such figures are already available in the Kikuchi et al (2012, 2021) and Lee et al (2013) references, to which the reader is directed in the introduction. No change has been made here.

3. In Line 64. The "symmetric" feature also exists in part of the TCWV. So it is not plausible enough to be considered a "difference".
We agree with the reviewer here, and have revised the analysis accordingly: "We also note the effect on both vorticity and TCWV is almost equal and opposite when three BSISO phases apart, even though the effect is much weaker on the latter. For example, the region of enhanced vorticity south of India and Sri Lanka associated with phase 2 is similarly suppressed in phase 5 – and thus the circulation response to `active' and 'suppressed' BSISO phases is approximately symmetric"

Minor opinions:

1. A typo at Line 59, should be Fig.3.
Corrected. Thank you.

2. It should be marked when the first time the abbreviation "TCWV" shows up.
Agreed, we have made this change.

3. Please write the full name of BSISO in the title.
We can make this change, but we note that 16 papers have been published with the abbreviation in the title, including in GRL (doi:10.1029/2018GL078321) and the Journal of Climate (10.1175/JCLI-D-20-0308.1). We defer to the Editor's preference here according to journal style.

4. Please write the full name of ERA5 in the subtitle in Line 90
We have replaced this subtitle with: "ERA5: the fifth generation ECMWF global reanalysis"

5. When you say the LPS genesis is half a phase behind the vorticity maxima from Line 140 to 145, what do you mean by "half a phase". Can you do a lag regression to show that? Show the order of water vapor, vorticity and LPS genesis reaching their maxima using lag regressions or any better method.
We have added a footnote in the revised manuscript to explain what we mean by fractional phases: "Recall that BSISO phase numbers are discretised from a continuous circular distribution, with e.g., 'phase 3' including phase numbers between 2.5 and 3.5. In the discrete space used in our maps, half a phase is the difference between being in the spatial centre of a phase and being at its boundary."

To answer the second part of the reviewer's comment, we further leverage these fractional – or continuous – phases. For reference, we provide a comparison between continuous phases and the standard eight discretised phases in Fig R3.5 below.

[Figure]

*Figure R3.14*

We now choose an appropriate sampling region – here using a 4°x4° box centred over Sri Lanka, following the text in L140-145, and making sure the region is large enough to reduce noise but small enough that it distinctly captures phase passages. We then take mean TCWV, relative vorticity, and precipitation over this box, using LOWESS smoothing to show them as functions of continuous BSISO phase in Figure R3.6. In addition, we plot a Gaussian kernel density estimate of LPS genesis rate (black). Vorticity leads, peaking at about 1.0; followed by LPS genesis, peaking at about 1.5; followed later by precipitation and TCWV, each peaking between 2.5 and 3.0. For brevity, we do not include these figures or analysis in the revised paper, but we do add a sentence summarising them at the end of the lines in question: "This pattern is confirmed through composite lag analysis using continuous BSISO phases (not shown)."

[Figure]

*Figure R3.15*

6. A question: why do you use relative latitudes and longitudes instead of the absolute ones in most of your plots? If you would like to state the topologies and locations, the absolute ones would be more helpful.
Use of relative latitude and longitude (or geodesic) coordinates is conventional for storm-centred composites, e.g. for tropical cyclones (doi:10.1175/1520-0469(1981)038<1132:OAOTCF>2.0.CO;2), extratropical cyclones (doi:10.1175/MWR3082.1), and even Rossby waves (doi: doi.org/10.1175/MWR-D-12-00012.1).

Neither of the ways of presenting absolute lons/lats possess a clear advantage. We could either take absolute in the literal sense, making an Earth-relative composite instead of a storm-relative composite. Given the relative footprint of LPSs is considerably smaller than the region in which they can be found, this would result in a smearing out of dynamical signals. Alternatively, we could simply add the mean LPS longitude and latitude to create a hybrid "absolute" coordinate, still storm-centred but with the origin shifted to (84°E, 19°N). But this would be misleading – a reader may then construe a point marked at (85°E, 27°N) to be in the Himalayan foothills, whereas in fact it is only that point convoluted by the track density function which may actually be as far south as 13°N (south India) or as far north as 35°N (Tibetan Plateau). For these reasons, no change has been made here.

I'm looking forward to seeing your response to the abovementioned questions.

---

## Referee Report (RR1)

Thanks for the quick response from the authors. I'm satisfied with the extra explanations. The paper is acceptable if all the adjustments can be made in the latest version. I'm looking forward to seeing the revised manuscript.